# A Focus on Electromobility within Smart City Solutions—Charging Stations, Renewable Energy, and Air Quality Monitoring

**DOI:** 10.3390/s22207841

**Published:** 2022-10-15

**Authors:** Radek Zavorka, Martin Paar

**Affiliations:** 1Department of Radioelectronics, Faculty of Electrical Engineering and Communication, Brno University of Technology, Technicka 12, 616 00 Brno, Czech Republic; 2Department of Electrical Engineering, Faculty of Military Technology, University of Defence, Kounicova 156/65, 662 10 Brno, Czech Republic; 3Deparment of Electrical Power Engineering, Faculty of Electrical Engineering and Communication, Brno University of Technology, Technicka 12, 616 00 Brno, Czech Republic

**Keywords:** electric vehicle chargers, battery, renewable energy, photovoltaic panels, air pollution monitoring

## Abstract

This paper reviews some of the existing methods for charging electric vehicles, generating renewable energy, and storing it. Plans of practical implementation in the city of Brno are compared with the situation in Glasgow. Moreover, it is essential to pay attention to integrated solutions in order to increase efficiency. Energy harvesting and charging systems are combined with an air quality measurement system and integrated into LED street lights. The collected data are sent to a central server for evaluation. The use of smart solutions is a modern approach to saving energy and reducing CO${}_{2}$ emissions in many sectors. As an example, the described solutions can be applied dually, in both civilian and military sectors. Considering the potential benefits of easier logistics or quiet operation, the potential military exploitation of technological capabilities is discussed from the perspective of enhancing citizens’ security and safety in cities.

## 1. Introduction

Within the context of the RUGGEDISED project, this paper examines the use cases of Glasgow, which is a Lighthouse city in the project, and Brno, a Fellow city. Among the solutions that Brno and Glasgow are currently working on, we will focus on supporting electromobility by installing electric vehicle (EV) chargers within additional technologies and in different city environments. For instance, saving energy is realized by using renewable energy sources. In particular, photovoltaic panels (PVs) will be installed on the roofs and walls of parking houses and industrial or residential buildings.

Renewable energy is an inevitable trend and an essential part of a sustainable lifestyle in the coming years. As shown in Figure 1, EV production is increasing [1], which means the development of charging infrastructure will be essential for EV adoption among the population. On the other hand, as electric energy consumption keeps growing, EVs will likely play a role in it. As a result, photovoltaic or other renewable sources for electricity production will be prioritized to meet the need for new resources to reduce fossil fuel consumption. Electric energy production, however, also depends on the available sources in the country and its duration in accordance with the load profile. Due to this, renewable energy is not the only solution; nuclear power and even gas-fired power plants are considered in some countries.

Several papers [2,3,4] focus on describing and designing EV chargers that use PV energy. The moment solar energy is transformed into electricity, it has to be stored in batteries or immediately sent to the grid or used for car charging. The production of EVs continues to grow [5], along with the battery capacity of EVs, requiring an increase in the number of chargers and chargers with sufficient power. The standard slow charger has a power output of 3.5–7.0 kW, which is adequate for recharging a car overnight or during working hours. In order to discover new opportunities, a solution implemented in Glasgow within the framework of RUGGEDISED uses fast chargers with a charging capacity of 50 kW, which is sufficient for short-term recharging [6]. An electric battery hub with a capacity of 250 kWh is located in the parking lot, where the chargers are situated. This battery could be charged to full capacity in one day in good solar conditions using energy from PV panels installed on the roof.

EV chargers could be integrated into lampposts which will also contain sensors for measuring air pollution [6]. The majority of streets have light poles that can be used for implementing these smart solutions. This approach makes it unnecessary to replace the current lighting poles with brand-new ones. Taking Glasgow as an example, where this solution is already being tested, several limitations have also been found. First, it is vital to adjust the cable connection to ensure that sufficient power is delivered. As a result, the solution is not suitable for fast charging, since Glasgow’s power capacity is only about 7 kW, which means that slow chargers are used. It is also necessary to move the poles to the side closer to the road, to prevent pedestrians from coming into contact with the power cable, which could cause injury. Additionally, near-field communication (NFC) readers are installed on the columns, allowing authorized users to charge their cars.

Monitoring air quality benefits citizens living in large cities. The sensors situated on the lamppost could measure pollutants such as dust particles (PM1, PM2.5, PM10), ozone (O${}_{3}$), nitrogen dioxide (NO${}_{2}$), nitric oxide (NO), carbon monoxide (CO), sulfur dioxide (SO${}_{2}$), hydrogen sulfide (H${}_{2}$S), etc. By the Council of the European Union Directive [7] (Annex III, Section C), sensors should be placed between 1.5 m (the breathing zone) and 4 m above ground level. It may be necessary to place the station at higher elevations (up to 8 m) in certain circumstances, for example, if it covers a large area. In addition to assessing air quality, long-term statistics could be generated, allowing a digital twin of the city to visualize air quality based on the analysis of these pollutants. Collected data could help us better understand the correlation between air quality and pollution from cars [8,9]. Regulation of high emissions vehicles entering the city center is possible, but controlling and managing this process might be challenging. Measured data from each lamppost could be transferred via IoT devices and a low-power wide-area network (LPWAN) such as LoRaWAN solutions and protocols, as described in [10,11].

Figure 2 illustrates the complexity of electromobility, which includes electric vehicles, charging stations, the use of renewable energy, as well as air quality measurement and assessment.

It is relevant to note that EVs emit no tailpipe emissions, thereby helping to reduce localized pollution, which is a significant issue in urban areas [12]. However, it does not mean EVs produce no emissions at all [13]. The production of electric vehicles emits more greenhouse gasses than that of combustion engine vehicles, particularly as the manufacturing of batteries is a highly demanding process [14]. Power generation, primarily through coal combustion, is one of the most polluting sectors, as are electric vehicles among other products that depend on this process [12]. Car production emissions must be taken into account. However, transportation fuels and vehicles significantly impact emissions in the operating phase. This approach is known as the well-to-wheel (WTW) method and can be divided into two sections: well-to-tank (WTT) and tank-to-wheel (TTW). By comparing the two approaches, one can estimate which phase has a more fundamental influence [15]. For internal combustion engine vehicles, it is TTW, while for EVs, it is energy production by process WTT. The WTT emissions analysis calculates the emissions derived from fuel extraction, refining, and distribution activities needed to fill the vehicle tank. TTW emissions are calculated by translating the power train energy efficiency of the vehicle into the environmental impact produced by fuel combustion to generate traction power. The environmental impact of internal combustion engine vehicles depends on the TTW phase [12]. In the future, pollution can be expected to be reduced in the WTT phase by changing the distribution of the different types of electricity sources and striving for maximum recyclability of both electric vehicles and battery systems.

According to a publication by the Federal Ministry for the Environment, Nature Conservation and Nuclear Safety, Germany [16], it is assumed that greenhouse gas production over the entire life cycle is lower for EVs compared with vehicles with internal combustion engines. It is around 30% less than petrol vehicles and around 23% less than diesel vehicles. However, several factors play a role in the quantitative assessment. The number of electric vehicles is growing, renewable energy sources are increasing, and emissions from combustion engine vehicles are decreasing. Yet, due to the limited number of EVs, it is difficult to evaluate their current effect on air quality. There is still much scope for future studies to monitor this effect and evaluate the correlation between EVs, air quality, and energy sources.

The power flows between EV batteries and the grid can be unidirectional or bidirectional [17]. All on-board EV chargers have implemented a unidirectional power flow. However, a more modern approach is to make use of bidirectional power control, which allows sending power from EV batteries into the grid [18]. As a result, grid stability can also be improved as peak demand can be covered. Nevertheless, the current batteries are limited by the number of charge–discharge cycles. Battery management should be based on active and reactive power [19]. The newly introduced operating modes are described in [20]. The first is home-to-vehicle operation (H2V), where the current of the EV charger is controlled according to the current consumption of electrical appliances in the household (this mode of operation is combined with G2V and V2G). In vehicle-for-grid (V4G), the EV battery charger is used to compensate for harmonics or reactive power, while simultaneously operating in G2V and V2G modes. Ref. [21] presents the design and implementation of a single-phase on-board bidirectional plug-in electric vehicle (PEV) charger. In addition to charging the vehicle’s battery, that system can provide reactive power support to the utility grid. A coordinated connection between plug-in EVs and the grid is necessary, as it may result in significant voltage deviations, power quality problems, overloads, etc. [22,23,24].

Electric vehicles are also very appealing to the military, but high reliability is a requirement. A benefit of this system is that it operates quietly, which could be a strategic advantage [25]. Additionally, fuel logistics are a complex process in large-scale military operations, so EVs may be a viable option owing to the widespread availability of electric power and the possibility of local production. Furthermore, energy management and monitoring of air pollutants are essential components of modern smart cities that could prove useful in cases of hostilities or disaster. Therefore, EV charging technologies being developed as part of smart cities may also be viewed from a military perspective. A separate chapter will be dedicated to army applications.

Using examples of implementation in Brno and Glasgow, this article explores well-established technologies related to electromobility, air quality sensing, and the effective applications of PV panels. Additionally, smart-city technologies are discussed from the perspective of dual use, which can enhance the safety and security of citizens. As such, it can be considered part of the paper’s original contribution.

The rest of the paper is organized as follows. Section 2 provides an overview of the current state of electromobility in the city of Brno and compares the situation with Glasgow. Section 3 presents an air quality monitoring system that allows control of the ratio of electric/combustion drives depending on the air pollution. Within Section 4, the described technologies are discussed from the point of view of dual use to protect citizens. The conclusion of the paper is provided in Section 5.

## 2. Electromobility in Brno and Glasgow

The growing number of electric vehicles is an important trend for modern cities. As the operation of cars uses energy that is not produced on site, this reduces air pollution. The main objective of this research is to assess the potential of renewable energy production and to determine the number and types of chargers for electric vehicles. The calculation of the annual energy production from photovoltaic power plants for specific areas is outlined.

### 2.1. Brno

Brno takes a responsible approach to electromobility, and many EV chargers have been built in the city in cooperation with several companies. A total of 70 charging stations were in place in August 2022 [26], with more than 100 planned by year’s end. Table 1 shows the list of companies, the power of chargers, and the number of charging points in the city. A vision is to build enough charging stations so that every resident is 5–10 min away from one. As a result, having access to charging stations will motivate citizens to acquire electric vehicles.

The city-owned company Teplarny Brno operates 26 charging stations currently, with plans to expand to 50 by 2022. Around 90% of these stations offer slow charging with 10–22 kW AC, and the charge time for EVs ranges between 2 and 8 h. In addition, three fast charging stations are located at traffic hubs, such as parking garages or large commercial buildings. The power of stations is between 22–150 kW and charging takes about 30–60 min, depending on the type of vehicle. Brno’s vision is to reduce emissions by 40% compared to the year 2000, and electromobility should be one of the critical factors. Therefore the goal of Brno is to build enough slow charging stations due to the basic assumption that most people will charge their cars at night or during working hours.

There is a growing trend toward electric vehicles in the Czech Republic as well as in Brno. Figure 3 and Figure 4 show progression in the number of all registered EVs in the time period from the year 2021 up to August 15. Data are provided by the Ministry of Transport of Czech Republic [27]. In both cases, the trend line is almost linear. Therefore, it is expected that the trend will be the same or grow faster in the next few years.

As already mentioned, producing energy from renewable sources is essential to meet the increased demand for electric vehicles, especially in residential areas of the city. In addition, Teplarny Brno, which operates tens of charging stations in Brno, has begun to integrate solar panels on the roofs of its buildings. Over time, the company intends to cover all of its buildings with PV panels. The first implementation can be found at Jihomoravske Namesti 1136/1a. There, a powerwall with a power of 2 × 22 kW/32 A is installed, along with solar panels on the roof of the technical building. Based on the roof capacity, the total covered area by photovoltaic panels is 156 m${}^{2}$. There are 78 PV panels with dimensions of 1 by 2 m. Each panel is expected to generate 0.4 kWp, bringing the overall result to 31.2 kWp. The building is also equipped with a storage battery equal to the capacity of a battery from a smaller EV, approximately 30 kWh. With ideal conditions, it could be charged in one day and the energy could be used as needed.

A map of solar irradiance in the territory of the city of Brno is available at the web portal datahub.brno.cz (accessed on 26 August 2022) [28,29] where it is possible to analyze the suitability of all building terms for PV installation. Based on the Brno 2019 digital surface model (stereophotogrammetric) with a resolution of one meter, the values of solar radiation intensity for the city of Brno have been calculated by the company TopGis, s.r.o. Calculations were made for the 365 days of the year within a time interval of half an hour. Figure 5 presents the illustrative analysis for the building of Teplarny Brno at Jihomoravske Namesti.

With the total area of the building being 432 m${}^{2}$, points with irradiance exceeding 1000 kWh/m${}^{2}$/year are highlighted with red color, which corresponds to 35 m${}^{2}$. With an average irradiance of 851.33 kWh/m${}^{2}$/year, the minimum and maximum values are 66.24 kWh/m${}^{2}$/year and 1077.56 kWh/m${}^{2}$/year, respectively.

From the gathered solar irradiance data it is possible to estimate the production of electric energy per year. There is a continuous increase in the efficiency of PV panels [30,31], but for the purpose of the experiment, ordinary panels with a 17% efficiency will be considered. Then it is possible to calculate the production of energy as: (1)$$E=I_{\mathrm{average}}\cdot A\cdot \eta =851.33\times 156\times 0.17=22577.27\;\mathrm{kWh}/\mathrm{year}$$
where $I_{\mathrm{average}}$ is the average irradiance per square meter per year, *A* is the area of PV panels, and η is the efficiency of used PV panels.

The European Commission provides a web application, the Photovoltaic Geographical Information System (PVGIS) [32], where it is possible to count theoretical energy gained from PV panels at a given address. Figure 6 is a graph generated by the PVGIS application showing how energy is spread out for every month. In both cases, energy production is about 22 MWh per year. The highest energy production takes place during the summer months, and the lowest production takes place during the winter months, as is to be expected in central European conditions.

### 2.2. Glasgow

A total of 292 charge points have been installed across 151 units in Glasgow as of April 2022 [33]. In collaboration with ChargePlace Scotland, the largest provider of public charging points across Scotland, 140 additional charging stations are scheduled for installation across Glasgow by the end of 2022, consisting of 7 kW and 22 kW charging stations [34,35].

The majority of current EV chargers are fast or rapid, which can charge most vehicles to 80% in about 30 min [33]. Table 2 provides details about the three types of chargers available in the city (fast, rapid, and slow). Additionally, a PV panel installation was carried out on the roof of the parking garage as part of the RUGGEDISED project; the power of the panels is 200 kWp, while the battery capacity is 200 kWh [36]. Glasgow City’s report shows [35] that more than 5300 drivers used the network for 43,112 sessions during 2021, consuming 988,000 kWh, equivalent to 2.8 million emission-free miles, avoiding 559.1 t of carbon dioxide. Hence, Scotland aims to phase out all petrol and diesel vehicles in Scotland by 2032 [35]. Almost half a million (447,000) battery electric vehicles (BEVs) are now driving on the United Kingdom’s (UK’s) roads [37].

Figure 7 shows a graph illustrating the progress of public EV charging points in the UK. It is evident that the trend is rapidly growing, and in comparison, between 2020 and 2021, there was a 36% increase in growth. In terms of power ratings, the four speeds tracked are described as slow (3–6 kW), fast (7–22 kW), rapid (25–99 kW), and ultra-rapid (100 kW+) [38].

## 3. Air Quality Monitoring

There are 13 air quality monitoring stations in Brno. Five stations are operated by Brno, six are owned by Czech Hydrometeorological Institute (CHMI), and two are by Public Health Institute. Several harmful substances, including NO${}_{2}$, NO, NOx, dust particles PM10, PM2.5, and PM1, as well as basic meteorological variables, such as pressure, humidity, temperature, wind speed, and direction, are measured. According to the World Health Organization (WHO), airborne particulate matter can cause significant health effects [39]. PM10 dust particles are particles with aerodynamic diameter < 10 μm [40] and they can settle in the bronchi and cause dangerous diseases [41,42]. A graph of one year of collected PM10 pollution data from Brno is shown in Figure 8. CHMI measured data and the locations of Brno-Uvoz (street in the city center) and Brno-Turany (airport on the outskirts of the city) were chosen to present the results.

Figure 8 shows concentrations of PM10 measured on the rolling 24 h average. Periodicity can be observed in all months of the year, although the concentration in the air increased slightly during winter, from January to February. The graph also indicates that the pollution level increased during the summer of 2022. There may be an increase in travel and production for companies compared to 2021 when the situation with the COVID-19 pandemic appeared to be more serious. Blank spaces in the graph are due to measurement errors.

A study such as this one suggests that air monitoring could be useful and that having more data would be beneficial for a healthy population. In addition, there is no need for new stands to integrate sensors into street LED lights. This approach is applied in Glasgow city, which could be replicated in Brno [6].

## 4. Smart Cities as Dual-Use Technology for Civil and Military Purposes

Smart cities are open domains for not only civil applications, but also for military purposes. With more than 56% of the world’s population and almost 75% of Europe’s population living in urban areas in 2020, the future potential importance of smart cities is on the rise [43]. Therefore, the dual use (for civil and military purposes) of the technology and infrastructure of smart cities should be at least considered during their conception and design. This should be able to help in times of crisis, natural disasters, or military conflicts. As the current large conflict in Ukraine, which was considered unlikely before 24 February 2022, shows, Europe is not without the risk of a classic major conflict and this risk should not be neglected. Smart cities include various technological fields, from sensors, and IoT (internet of things) up to energy management and smart metering, as can be seen in [44]. The number of sensors and their communication also opens the questions of energy consumption effectivity [45], cybersecurity [46], and in general, the resilience of future smart cities [47]. This article focuses on topics dealing with electric energy consumption in smart cities, local production of electricity with a specialization on EVs and their charging, and the monitoring of air quality.

In general, an army’s operation requires extensive logistics. Currently, opportunities to cover part of its energy consumption using local sources are being investigated. This could decrease the vulnerability of logistics and potential losses of troops and equipment. Local production of electric energy without fuel delivery requirements is generally accomplished by photovoltaic or wind power sources. These sources are weather dependent and the energy gained is usually insufficient to fully satisfy the army’s needs. However, with a connection to battery systems or fuel cell technology, these sources can offer a solution to fulfill at least part of the army’s energy consumption. The second advantage of this approach, especially with photovoltaics, is their silent operation.

The military’s electricity consumption ranges from supplying bases to vehicles to equipping soldiers. Wearable and portable soldier equipment contains various gear such as a GPS, personal radio, and mini-UAVs. For example, for a 72 h mission, a soldier in Afghanistan in 2012 required 618 Wh of energy and 7 kg of batteries [48]. The ability to fuel equipment without the requirement of supply logistics could improve the usability of the equipment as well.

The next area of energy consumption is electromobility. Electromobility, in particular EVs, is slowly becoming part of civil transportation. The military, however, has higher requirements on their vehicles [25], which leads to delayed usage in comparison to civil use. The currently used military electromobility is focused on electric bikes or electric motorcycles that are quieter and are used for scouting (e.g., Australia) [49] or for hit-and-run or shoot-and-scoot tactics (e.g., Ukraine) [50]. Another application of electric propulsion is for small support vehicles, ranging from drones up to partly autonomous vehicles, such as the Themis [51], which were newly introduced in several European armies. In particular, heavier vehicles not only use electric batteries but also have an extended range from gas- or diesel-powered generators. It means that they are not BEVs (battery electric vehicles) but HEV (hybrid electric vehicles). With the increase in battery capacities and new battery materials, the next step will lead to light utility military vehicles. In this area, non-military vehicles, such as Bollinger prototype cars [52], could show the advantages of the electric vehicle concept. The current developments in the area of electromobility will enable this type of vehicle to be used in armies in the near future. For the development of heavy vehicles, such as heavy tracks [53], tanks, or tracked artillery, the current focus is more on hydrogen technology, where the battery system will be used as well.

Fueling military equipment and vehicles with electric energy would decrease the number of fuel types. Electric energy can be produced from different types of primary energy sources, thus decreasing the logistic complexity, which is usually available from an electric grid outside of the disaster area or in the background of the combat area. In a disaster or combat area, the production of electric energy can be secured by classic generators or by renewable sources, as electric energy delivered through the electric grid is usually vulnerable to the effects of the disaster or military operations. A similar approach can be taken with smart cities outside of disaster or war-affected zones. A combination of renewables and an electrical grid can be used, which can also improve the reliability of the energy supply in times of crisis. In the disaster/combat zone, a certain amount of energy should be able to be produced by renewable sources that operate as independent microgrids (the ability depends on technical design). It could provide at least some energy for soldiers and civilians, although it would depend on the weather or intensity of combat operations. For a short time, the battery systems or electric vehicles in V2G (vehicle-to-grid), V2H (vehicle-to-home), or V2L (vehicle-to-load) modes could be used as an energy source as well.

Section 3 of the article focuses on air quality evaluation. This is one of the information feeds that can be utilized by the military as the information flow is provided by smart cities. The sensors in the city itself can bring more information about enemy activities or civil population movements. For this purpose, the sensors used should be able to operate and provide information for a specified time without an external power supply or at least with basic resilience to enemy activities (e.g., keep data integrity). An independent power supply should also be a requirement for the use of these sensors in case of a natural disaster.

The infrastructure of smart cities can be considered as dual-use technology (civil and military) and should be able to help in times of disaster, security incidents, warfare activity (e.g., bombing), or situations when the city is in a combat area. Additionally, the infrastructure could provide better conditions for civilians and for military or security forces in times of crisis by providing independent energy sources and information flow from the area, which can assist in improving surveillance.

## 5. Conclusions

The article investigated the current state of the EV market and the latest trends. Data show a positive increase in the number of electric vehicles registered in the Czech Republic, as well as in the city of Brno. Approximately 19 times more electric vehicles are registered in the United Kingdom than in the Czech Republic; however, the larger population and territory in the UK must be taken into account. Using electric vehicles effectively meets the population’s transportation needs without adversely affecting air quality. Combined with renewable energy sources, they hold a significant influence when it comes to charging infrastructure and accommodating the demands of EVs. In particular, photovoltaic panels appear promising, and scientific advancements ensure increased efficiency in energy conversion. The possibility of calculating electricity production by PV panels for a given location is outlined. A detailed analysis of the city of Brno regarding the availability of EV chargers was carried out, evaluated, and compared with Glasgow. Moreover, the possibility of air quality monitoring in Brno was described, and selected locations were presented and compared.

The military’s perspective on smart cities and electromobility has also been reviewed and discussed. Due to the possibility of silent operation and ease of fuel distribution, EVs are an appealing approach for a military area. Although a smart city can utilize its sensors to improve its military’s knowledge of the situation, there is a risk that an enemy can use this information for their own benefit. As a result, smart cities with sensor systems have potential benefits and drawbacks that must be considered in developing them.

## Figures and Tables

**Figure 1 sensors-22-07841-f001:**
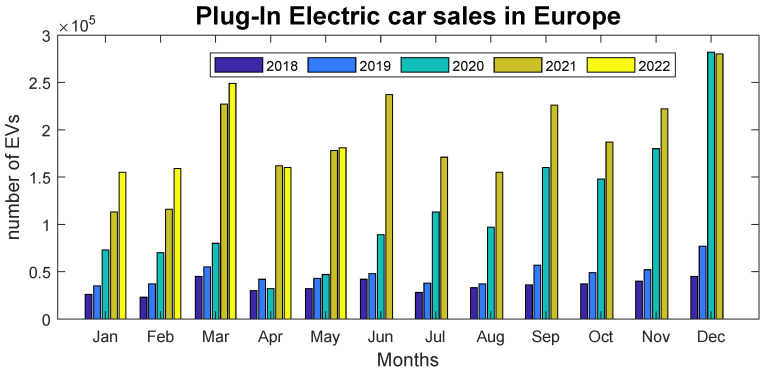
Production of EVs in Europe [1].

**Figure 2 sensors-22-07841-f002:**
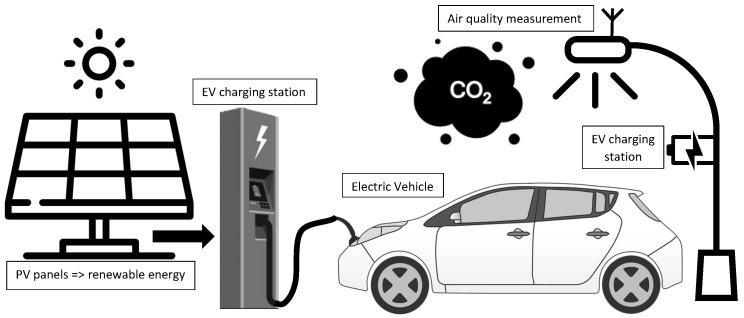
Application of electromobility.

**Figure 3 sensors-22-07841-f003:**
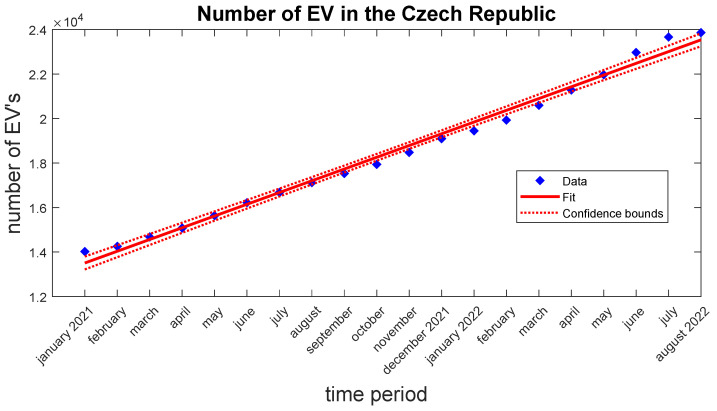
Graph of the registered EVs in the Czech Republic.

**Figure 4 sensors-22-07841-f004:**
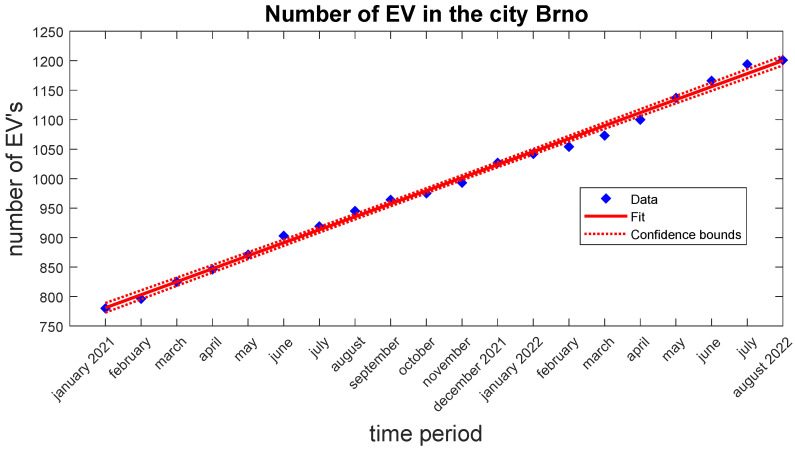
Graph of the registered EVs in Brno.

**Figure 5 sensors-22-07841-f005:**
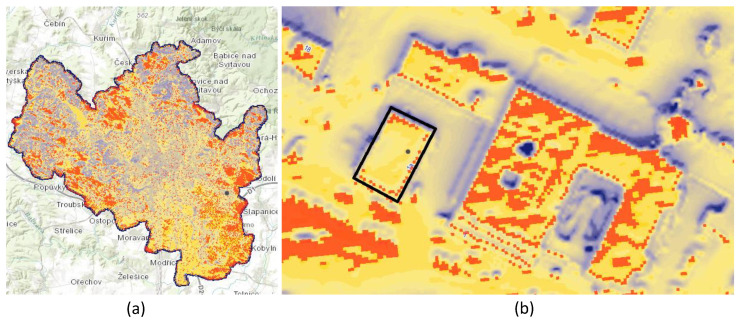
Maps of solar irradiance, (**a**) area of Brno and (**b**) zoom to address Jihomoravske Namesti 1136/1a [28,29].

**Figure 6 sensors-22-07841-f006:**
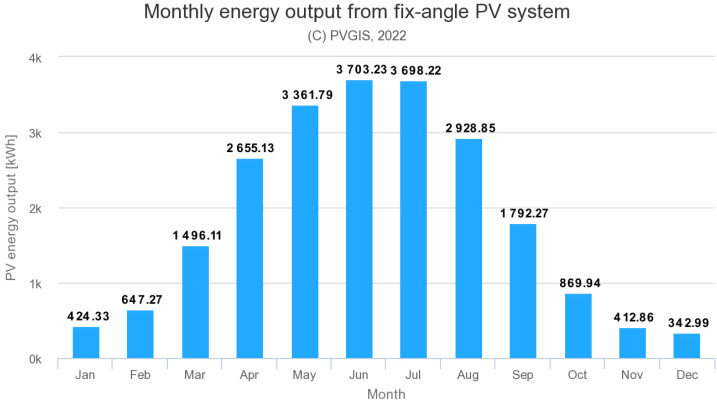
Monthly energy output of PV system calculated by the PVGIS application [32].

**Figure 7 sensors-22-07841-f007:**
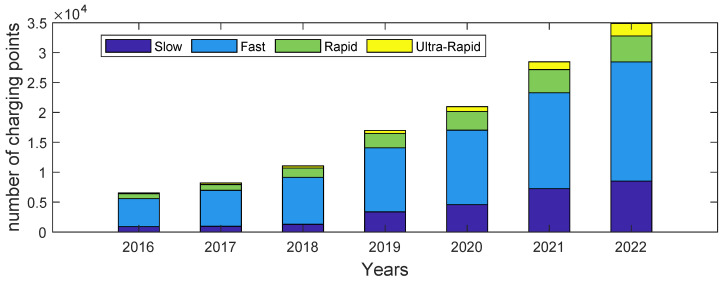
Number of public charging points in the UK [38].

**Figure 8 sensors-22-07841-f008:**
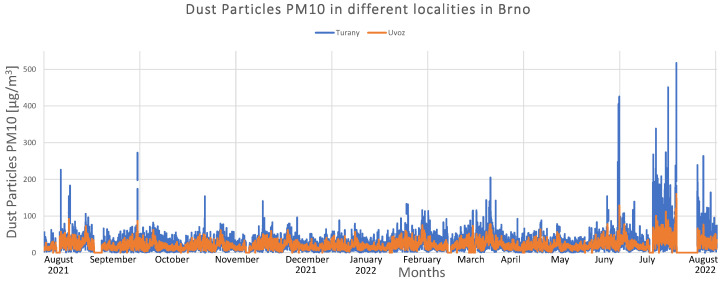
P10 pollutant monitored in Brno.

**Table 1 sensors-22-07841-t001:** Parameters of EV chargers in Brno.

Owners	Power of Chargers (kW)	Number of Chargers
Teplarny Brno	22–150	26
CEZ	11–50	17
Tesla	14–22	5
E.ON	11–50	9
PRE	22–150	8
Olife Energy	40	1
Others	-	±4

**Table 2 sensors-22-07841-t002:** Parameters of EV chargers in Glasgow [33].

Charger Type	Aproximate Charge Duration	For Use When...
7 kW “Slow”	6–8 h	it is planned to stay longer and a quick charge is not required or when just a top-up is required. This slower charge is also thrifty on the battery.
22 kW “Fast”	2–4 h	it is a short stay and requires a substantial charge.
50 kW “Rapid”	<1 h	the situation requires a quick top-up to allow one to continue a journey or to get travelers to their destination. Rapid charges can cause that battery to deteriorate over time, resulting in a reduced capacity.

## Data Availability

Not applicable.

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
