# Peer review of "A Focus on Electromobility within Smart City Solutions—Charging Stations, Renewable Energy, and Air Quality Monitoring"

_sensors, 2022, doi:10.3390/s22207841_

Round 1
Reviewer 1 Report
This paper explores well-established technologies related to electromobility, air quality sensing, and the effective applications of PV panels by using examples of implementation in Brno and Glasgow. This is an interesting study and its quality is good. In my opinion, the paper can be accepted after minor corrections.
1. Authors investigated the current state of the EV market and the latest trends. While qualitative conclusion may be good, there’s also the need for quantitative to close the work in this study. It would be good if the authors could add some quantitative analysis to study the relationship between charging stations, renewable energy, and air quality monitoring.
2. Figure 1: I think it is more appropriate to display the changes in the production of electric vehicles in a time series order.
3. Equation (1) should be justified further.
4. There are too many long sentences in this paper, which are not clear enough. Authors need to sort out the long sentences in the paper.
Author Response
1.Added a paragraph responding to the quantitative evaluation of the results.
According to a publication by the Federal Ministry for the Environment, Nature Conservation and Nuclear Safety, Germany [16], it is assumed that greenhouse gas production over the entire life cycle is lower for EVs compared with vehicles with internal combustion engines. It is around 30 % less than petrol vehicles and around 23 % less than diesel vehicles. However, several factors play a role in the quantitative assessment. The number of electric vehicles is growing, renewable energy sources are increasing, and emissions from combustion engine vehicles are decreasing. Yet, due to the limited number of EVs, it is difficult to evaluate their current effect on air quality. There is still much scope for future studies to monitor this effect and evaluate the correlation between EVs, air quality, and energy sources.
- Maybe it could be good to use Figure 1, graph of EV production, in time series, but I don't have raw data and if I will create my own graph there will be inaccuracies. So in my opinion using the original chart is a better solution in this case.
- Equation 1 was explained and justified.
- Language correction was applied along all text. Long sentences were corrected.
Reviewer 2 Report
This paper based on the examples of implementation in Brno and Glasgow explores well-established technologies related to electromobility, air quality sensing, and the effective applications of PV panels. Additionally, the discussion of smart-city technologies from the perspective of dual use, which can enhance the safety and security of citizens, can be considered an original contribution of the paper. So the article represents the research in hot and essential topic, that suites to Sensor journal scope. SSelection of special issue " Sensor Applications for Military* and Public Use: The Duality into Action" is correct. The novelty is acceptable. Thus my general assessment of the paper is positive.
Some recommendations to improve paper:
-> the results presented in abstract should be quantified. Of course only the main results.
-> I recommend including a chart that will discuss the application of electromobility that is discussed in this paper. I think a good place to locate it is section 1.
-> Between 2. Electromobility in Brno city and Glasgow and 2.1. Brno city please add a general introduction to section 2.
-> For figure 7 "Pollutant P10 monitored in Brno" please delate the borderline.
Author Response
1.Added a paragraph responding to the quantitative evaluation of the results.
According to a publication by the Federal Ministry for the Environment, Nature Conservation and Nuclear Safety, Germany [16], it is assumed that greenhouse gas production over the entire life cycle is lower for EVs compared with vehicles with internal combustion engines. It is around 30 % less than petrol vehicles and around 23 % less than diesel vehicles. However, several factors play a role in the quantitative assessment. The number of electric vehicles is growing, renewable energy sources are increasing, and emissions from combustion engine vehicles are decreasing. Yet, due to the limited number of EVs, it is difficult to evaluate their current effect on air quality. There is still much scope for future studies to monitor this effect and evaluate the correlation between EVs, air quality, and energy sources.
- Schematic figure (Figure 2) discussed application of electromobility and paragraph with comments were added.
Figure 2 illustrates the complexity of electromobility, which includes electric vehicles, charging stations, the use of renewable energy, as well as air quality measurement and assessment.
- General introduction of chapter 2 was added.
The growing number of electric vehicles is an important trend for modern cities. As the operation of cars uses energy that is not produced on site, this reduces air pollution. The main objective of this research is to assess the potential of renewable energy production and to determine the number and types of chargers for electric vehicles. The calculation of the annual energy production from photovoltaic power plants for specific areas is outlined.
- Borderline around figure (Figure 8) "Pollutant P10..." was deleted.
Round 2
Reviewer 2 Report
I recommend to publish this paper in present form.